# An Optimal Resonant Frequency Band Feature Extraction Method Based on Empirical Wavelet Transform

**DOI:** 10.3390/e21020135

**Published:** 2019-02-01

**Authors:** Zezhong Feng, Jun Ma, Xiaodong Wang, Jiande Wu, Chengjiang Zhou

**Affiliations:** 1Faculty of Information Engineering and Automation, Kunming University of Science and Technology, Kunming 650500, China; 2Engineering Research Center for Mineral Pipeline Transportation of Yunnan Province, Kunming 650500, China

**Keywords:** empirical wavelet transform, scale-space histogram, spectral negentropy, correlation coefficient, rotating machinery

## Abstract

The Empirical Wavelet Transform (EWT), which has a reliable mathematical derivation process and can adaptively decompose signals, has been widely used in mechanical applications, EEG, seismic detection and other fields. However, the EWT still faces the problem of how to optimally divide the Fourier spectrum during the application process. When there is noise interference in the analyzed signal, the parameterless scale-space histogram method will divide the spectrum into a variety of narrow bands, which will weaken or even fail to extract the fault modulation information. To accurately determine the optimal resonant demodulation frequency band, this paper proposes a method for applying Adaptive Average Spectral Negentropy (AASN) to EWT analysis (AEWT): Firstly, the spectrum is segmented by the parameterless clustering scale-space histogram method to obtain the corresponding empirical mode. Then, by comprehensively considering the Average Spectral Negentropy (ASN) index and correlation coefficient index on each mode, the correlation coefficient is used to adjust the ASN value of each mode, and the IMF with the highest value is used as the center frequency band of the fault information. Finally, a new resonant frequency band is reconstructed for the envelope demodulation analysis. The experimental results of different background noise intensities show that the proposed method can effectively detect the repetitive transients in the signal.

## 1. Introduction

Rolling bearings are the most widely used components in mechanical systems [1]. In actual operation, the rotating machinery is mostly in complex environments such as those with high speeds, high loads, a large temperature difference and so on, which causes the probability of failure to be much higher than for other equipment [2]. With the increasing requirements of high precision, high speed and large loads of mechanical equipment, it is becoming more and more complicated in terms of structure and larger in size, accompanied by complex abnormal noise [3]. As a result, it becomes more and more difficult to identify potential faults from the machine, which makes the fault feature extraction of rotating mechanical vibration signals a huge challenge [4].

In practice, time-frequency analysis is widely used to detect sensitive features in vibration and sound signals [5]. The Empirical Mode Decomposition (EMD) method, developed by Huang et al. [6], can adaptively decompose complex data sets into finite intrinsic modal functions (IMF) without getting any prior knowledge. Especially in the case of intermittent signals, impulse interferences or strong noise, the method presents some shortcomings such as mode mixing effect and endpoint effect when decomposing signals [7]. To solve this problem, various improved methods have emerged and have achieved some improvement of the results in various applications [8,9,10]. An approach called "empirical wavelet transform" (EWT), which combines the adaptability of EMD with the theoretical framework of wavelet analysis, effectively solves the mode mixing phenomenon problem [11]. By adaptively dividing the Fourier spectrum of signals, EWT constructs an appropriate orthogonal wavelet filter bank to extract the AM-FM signal components with a compact support set of spectra [12]. Since each IMF has a limited bandwidth with different central frequencies, it is important to optimally segment the Fourier spectrum in this method. To solve the problem of an automatic division of the Fourier spectrum in the EWT, Wang et al. proposed a new boundary segmentation method based on sparsity guidance [13]. First the characteristic frequencies of the extracted faults are calculated, and then applied to the sparsity function to find the cutoff frequencies and divide the spectrum boundaries. Xu [14] constructed the trend components by analyzing the waveform characteristics of the spectrum. Then the minimum value of the trend component was used to divide the boundary, and a fast empirical wavelet method was proposed. Yu [15] took the Harmonic Significance Index as the optimization object, and extracted the fault characteristics of bearing by quantitatively analyzing the sensitive frequency band. These methods solve the problem of spectrum segmentation by studying the characteristics of the Fourier spectrum. In order to achieve self-adaptability, Gilles applied the parameterless clustering scale-space histogram method to the EWT, which can achieve spectrum segmentation with a data-driven approach [16]. However, when the signal-to-noise ratio (SNR) of the signal is relatively low, the spectrum will be seriously disturbed, and the decomposition results will show a lot of noise. Pan noted through an in-depth study of the method that when too fine of a frequency partition is set, many narrow band modes are extracted with two corresponding problems. On the one hand, several modes may show the same modulation information, resulting in unnecessary redundancy. On the other hand, too coarse of a frequency partition will mix different mono-components and may introduce artifacts [17]. To compensate for the over-decomposition of vibration modes, Pan used priori knowledge to set the scale parameter as the predetermined value, aimed at solving the problem of signal distortion caused by the excessive frequency band. This approach provides a way to apply scale-space segmentation in a noisy background. At the same time, Pan also noted in their work that the proposed method did not give recommendations for the scale-space parameters under different noise intensities. 

The above literature shows that the improvement of EWT mostly focuses on how to optimize the division of the spectrum. Since the randomness of noise is reflected as the uncertainty of spectrum division, it is difficult to construct the optimal filter only by studying the spectral features. Therefore, by effectively analyzing the characteristics carried by each empirical mode, the optimal resonance band constructed by the data-driven method will have better universality and stronger robustness to noise. In the event of a local failure in a rolling bearing, a periodic instantaneous impact is generated, and the bearing system is stimulated to resonate and decay exponentially [18]. Antoni proposed Spectral Kurtosis (SK) [19] and Fast Kurtogram (FK) [20] based on an envelope analysis to determine the center frequency and bandwidth of the optimal resonance band. Nevertheless, in the case of a low SNR, the center frequency obtained by the Kurtogram is inaccurate and the bandwidth is too wide. Meanwhile, it is a research hotspot in recent years to determine the center frequency position by fully considering the cyclostationality of fault signal. When the studied signal has a cyclostationary component, the square envelope spectrum will show a peak value at the specific cyclic frequency of the phenomenon [21]. Based on this characteristic, Tomasz et al. proposed a Protrugram method [22] based on the narrowband envelope spectrum kurtosis. Unfortunately, this method still has the problem that the bandwidth needs to be preset, and the ideal effect is not achieved when there is noise interference [23]. To capture the impulsiveness and cyclostationarity of the signal, Antoni et al. used Shannon entropy as a reflection of the transient deviation of the signal, equated the negative entropy with kurtosis to measure the energy fluctuation in the spectrum, creatively combining Kurtogram with Protrugram, and proposed the average spectral Negentropy (ASN) method by measuring the negative entropy in square envelope and square envelope spectrums respectively, which is very important to precisely locate the fault frequency center of bearings [24]. One point worth improving is this method uses the average partitioning of each level for the segmentation of the spectrum, which is rough and lacks physical significance. Meanwhile, it can be observed that the precise location of the center frequency is the key to determine the optimal resonance frequency band. After consulting, we found that the combination of entropy and decomposition methods has been extensively used for noise elimination, evaluation of mechanical operation state, identification of fault types and so on. For example, Wu et al. applied transfer entropy to multivariate empirical mode decomposition to improve the robustness of detection while mitigating the mode mixing phenomenon [25]. Li uses the multiscale-dispersion entropy to identify the noise IMFs decomposed by the complete ensemble empirical mode, which plays a role in de-noising [26]. Li et al. applied the improved time-frequency entropy to EWT by constructing a classifier to identify the type of fault in the rolling bearing [27]. Inspired by the above literature, we firstly apply the idea of combining entropy with decomposition method to the task of accurately constructing the optimal fault frequency band. It is worth noting that in actual operating conditions, abnormal impulse interference is often accompanied by strong randomness, and the increase of entropy may not always be associated with an increase in dynamic complexity, which may present multiple high values on different frequency bands, and it is still difficult to determine the optimal fault center location [28,29]. Therefore, how to improve the ASN part of the proposed method to make it more accurate and robust is another core issue of this paper. 

In this paper, an optimal resonant frequency band feature extraction method based on EWT is proposed. First, the Fourier spectrum of vibration signals is segmented by a parameterless clustering scale-space histogram to obtain all the empirical modes. What’s more, in order to improve the accuracy and robustness of the method to select the center frequency in the face of different intensity noise, the correlation coefficient is used as the adaptive regulatory factor to the ASN. Finally, by effectively combining the adjacent modes with the central mode, the optimal resonant frequency bandwidth is established for envelope spectrum analysis.

The remainder of this paper is organized as follows: In Section 2, the basic theory of the EWT and scale space representation is reviewed. In Section 3, the specific implementation process of the AEWT method in repetitive transients is given. The simulation signals are studied using the proposed method in Section 4, and the actual application performance of the proposed method is shown in Section 5. Finally, the discussion and some conclusions are presented in Section 6.

## 2. Theoretical Background

### 2.1. Empirical Wavelet Transform

The EWT is discussed under the Shannon-Nyquist criterion. First, it is assumed that the Fourier spectrum range is normalized to [0, π]; then, the signal of the Fourier spectrum adaptive segmentation is set to represent different modes of *N* interval, and the required number of boundaries is *N* + 1, which is constructed as a set of filter banks, where each segment can be recorded as $\Lambda_{n}=[\omega_{n-1},\omega_{n}]$, where $\omega_{0}=0$ and $\omega_{n}=\pi$, i.e., $\sum_{n=1}^{N}\Lambda_{n}=[0,\pi ]$. Thus, an empirical wavelet function with orthogonal properties is obtained. Finally, the AM-FM single component with an orthogonality property of the Fourier spectrum can be extracted by a wavelet transformation of the segmented interval in the frequency domain.

After determining the segmentation interval Λn, the empirical wavelet is defined as the bandpass filter on each Λn. Gilles et al. adopts the construction method of Meyer’s wavelet. The empirical scale function ${\hat{\phi }}_{n}$ and the empirical wavelet ${\hat{\psi }}_{n}$ are respectively defined as follows:
(1)$${\hat{\phi }}_{n}(\omega )=\{\begin{array}{c} 1\\ \cos\left[\frac{\pi }{2}\beta \left(\frac{1}{2\gamma \omega_{n}}\left(|\omega |-(1-\gamma )\omega_{n}\right)\right)\right]\\ 0 \end{array}\;\begin{array}{c} \mathrm{if}|\omega |\leq (1-\gamma )\omega_{n}\\ \mathrm{if}\;(1-\gamma )\omega_{n}\leq |\omega |\leq (1+\gamma )\omega_{n}\\ \mathrm{otherwise} \end{array}$$
(2)$${\hat{\psi }}_{n}=\{\begin{array}{c} 1\\ \cos\left[\frac{\pi }{2}\beta \left(\frac{1}{2\gamma \omega_{n+1}}\left(|\omega |-(1-\gamma )\omega_{n+1}\right)\right)\right]\\ \sin\left[\frac{\pi }{2}\beta \left(\frac{1}{2\gamma \omega_{n}}\left(|\omega |-(1-\gamma )\omega_{n}\right)\right)\right]\\ 0 \end{array}\;\begin{array}{c} \mathrm{if}\;(1+\gamma )\omega_{n}\leq |\omega |\leq (1-\gamma )\omega_{n+1}\\ \mathrm{if}\;(1-\gamma )\omega_{n+1}\leq |\omega |\leq (1+\gamma )\omega_{n+1}\\ \mathrm{if}\;(1-\gamma )\omega_{n}\leq |\omega |\leq (1+\gamma )\omega_{n}\\ \mathrm{otherwise} \end{array}$$


In Equations (1) and (2), *ω* represents the angular frequency, where the function $\beta (x)\in C^{k}([0,1])$ is defined as follows:
(3)$$\beta (x)=x^{4}(35-84+70x^{2}-20x^{3})$$


When $\gamma <\min_{n}\left(\frac{\omega_{n+1}-\omega_{n}}{\omega_{n+1}+\omega_{n}}\right)$, $\left[{\hat{\phi }}_{1},{\left\{{\hat{\psi }}_{n}\right\}}_{n=1}^{N}\right]$ can satisfy the requirements of a tight frame in the $L^{2}(R)$ space. An empirical wavelet filter bank is constructed by applying Fourier spectral segment information to Equations (1) and (2), and then the empirical wavelet transform can be defined by using a similar traditional wavelet transform. The detail coefficients are derived from empirical wavelet functions and signals. The inner product is given by:
(4)$$W_{x}(0,t)=\langle f,\phi_{n}\rangle =\int x(\tau )\bar{\phi_{n}(\tau -t)}d\tau$$


The detail coefficient $W_{x}(n,t)$ is defined as the inner product of the detection signal *f* and the wavelet scaling function, given by:
(5)$$W_{x}(n,t)=\langle f,\psi_{n}\rangle =\int x(\tau )\bar{\psi_{n}(\tau -t)}d\tau$$


The extracted modes *f_k_* is given by:
(6)$$\begin{array}{l} f_{0}(t)=W_{x}(0,t)*\phi_{1}(t)\\ f_{k}(t)=W_{x}(k,t)*\psi_{k}(t) \end{array}$$


The detailed algorithm of the EWT can be found in [12].

### 2.2. Scale Space Histogram

In summary, the scale space approach introduces a parameter in the data analysis and defines the size of the parameter. Then, the data information corresponding to different parameters is obtained through constantly changing scale parameters, and the intrinsic characteristics of the data information are excavated. Witkin [30] and Lindeberg [31] noted that the discrete Fourier transform *X*(*f*) of the finite-length discrete-time vibration signal *x*(*t*) can be filtered by a kernel function, and some parameters of the kernel function are used as a scale, whose corresponding discrete scale-space representation *L*(*x*,*t*) is defined as the following equation:
(7)$$L(x,t)=\sum_{n=-\infty }^{+\infty }g(n;t)⊗X(f)$$
where ⊗ denotes the convolution product; *g*(*n*;*t*) denotes the Gauss kernel function, given by Equation (8). and the parameter *t* in the function is the scale, which means that we can obtain *L*(*x*,*t*) on different scales by an iterative convolution:
(8)$$g(n;t)=\frac{1}{\sqrt{2\pi t}}e^{-n^{2}/(2t)}$$


In practice, in order to have a finite impulse response, the range of the filtering is reduced from −∞–+∞ to −*M*–+*M*. When *M* is sufficiently large, the approximate error of the Gauss is negligible. A common choice is to set *M* to 3 < C < 6 (which means that the size of the filter increases relative to *t*). This paper determines C = 6, to ensure an approximation error of less than 10^−9^. The sampling of *t* is conducted as $\sqrt{t}=s\sqrt{t_{0}}$, where $s=1,\ldots ,s_{\max}$, and $\sqrt{t_{0}}=0.5$ represents half of the distance between the two samples. Because we use a finite length signal, there is no interest in studying more than $\sqrt{t_{\max}}=x_{\max}$, which means $s_{\max}=2x_{\max}$.

The scale-space representation of the histogram gives a key property, where the number of minimum values of *L*(*x*,*t*) relative to *x* is a decreasing function of *t* and *s*. Therefore, each local minimum produces a scale-space curve of different lengths. Since the scale-space feature between two local minimum values in the histogram has a sufficient length, the distinguishing meaningful empirical modes are quantified to find the scale-space feature whose length is longer than the threshold *T* [16]. The remaining problem is to develop an adaptive method to determine the appropriate value of *T* for different signals. This can be defined as a two-coalescing class problem on a collection. In this paper, *k*-means clustering is selected to determine the appropriate value of *T*.

## 3. Optimal Resonant Frequency Band is Established Based on the EWT

When the signal contains noise and impulse interference, parameterless clustering scale-space histogram method will divide various narrowbands on Fourier spectrum. Our strategy is to determine the center position and then construct the optimal filter size to qualitatively detect the fault characteristics. The detailed process of our approach will be described in the following subsections.

### 3.1. Average Spectral Negentropy Operator

Spectral kurtosis, as a statistical tool, can be used to detect the anomaly of the system by its sensitivity to singular signals. It is defined in the frequency band according to the spectral cumulant as follows:
(9)$$K_{x}=\frac{S_{4x}(IMF_{i})}{S_{2x}^{2}(IMF_{i})}-2=\frac{\langle \left|SE_{x}(n;IMF_{i}{)}^{4}\right|\rangle }{\langle \left|SE_{x}(n;IMF_{i}{)}^{22}\right|\rangle }-2$$
where $S_{4x}(IMF_{i})$ is the fourth-order spectral cumulant, $S_{2x}^{2}(IMF_{i})$ is the 2*n*-order instantaneous moment. 〈⋅〉 is the mean operation, and the discrete signal *x*(*n*) of length *n* is defined, whose squared envelope (SE) on the *i*-th mode is defined as follows:
(10)$$SE_{x}(n;IMF_{i})={\left|x(n;IMF_{i})+jH((n;IMF_{i}))\right|}^{2}$$
where *H*(⋅) is the Hilbert transform. Fault pulse in a mechanical system causes a sudden change in the vibration signal, it means that the entropy value of the system also changes. When the bearing is in a normal state, the energy fluctuation of the signal is constant and the spectral entropy value reaches a maximum. Conversely, when the fault pulse causes the energy fluctuation to change, the spectral entropy is at a minimum, which is the opposite of the kurtosis index. To maintain the same physical significance as the spectral kurtosis for detecting the transient change of the system, the Spectral Negentropy of the time domain was taken to characterize the pulse characteristics on all modes, defined as follows:
(11)$$\Delta I_{e}(IMF_{i})=\langle \frac{SE_{x}(n;IMF_{i}{)}^{2}}{\langle SE_{x}(n;IMF_{i}{)}^{2}\rangle }\ln(\frac{SE_{x}(n;IMF_{i}{)}^{2}}{\langle SE_{x}(n;IMF_{i}{)}^{2}\rangle })\rangle$$


As can be seen, the Spectral Negentropy in the time domain can be regarded as the spectral kurtosis with a weight of $\ln(SE_{x}(n;IMF_{i}{)}^{2}/\langle SE_{x}(n;IMF_{i}{)}^{2}\rangle )$. However, due to the influence of noise, only the pulse characteristics will have large errors, the cyclostationarity characteristics shown when a local faults exist should also be taken into account at the same time. Periodic pulse faults appear as a series of repeated transients in the time domain (i.e., harmonics in the square envelope spectrum). Therefore, the spectral negative entropy in the frequency domain is used to reflect the cyclostationary characteristics of mode *i*, defined as follows:
(12)$$\Delta I_{E}(IMF_{i})=\langle \frac{SES_{x}(\alpha ;IMF_{i}{)}^{2}}{\langle SES_{x}(\alpha ;IMF_{i}{)}^{2}\rangle }\ln(\frac{SES_{x}(\alpha ;IMF_{i}{)}^{2}}{\langle SES_{x}(\alpha ;IMF_{i}{)}^{2}\rangle })\rangle$$


In the formula: $SES_{x}(\alpha ;IMF_{i})$ is the square envelope spectrum (SES) on the *i*-th mode, defined as:
(13)$$SES_{x}(\alpha ;IMF_{i})=F(SE_{x}(n;IMF_{i})$$


In this formula *F* is the Fourier transform and α is the frequency change value. When the transient fault pulse exists in the bearing, both the values of $\Delta I_{e}(IMF_{i})$ and $\Delta I_{E}(IMF_{i})$ increase. Combining them can simultaneously measure both impulsiveness and cyclostationarity. Then, the expression of the average spectral negentropy (ASN) of the *i*-th mode is as follows:
(14)$$\Delta I_{1/2}(IMF_{i})=\frac{\Delta I_{e}(IMF_{i})}{2}+\frac{\Delta I_{E}(IMF_{i})}{2}$$


### 3.2. ASN Modified by Correlation Coefficient

In practice, when the ASN was used as the evaluation index selected by the center, we found that the effect was not ideal. As described in [28], due to interaction and coupling effects between machine components, its high value will appear in multiple modes. In a large number of experiments, it was also found that when the detection signal is relatively pure, the ASN which relies more on cyclostationarity will have a larger error, so it should be adaptively corrected.

The correlation coefficient is used as a statistical indicator to reflect the close relationship between the variables. The high correlation model can show more characteristics with the original vibration signal *x*(*t*). The correlation coefficient is calculated as follows:
(15)$$\rho_{X,Y}=\frac{\mathrm{cov}(X,Y)}{\sigma_{x}\sigma_{y}}=\frac{E(XY)-E(X)E(Y)}{\sqrt{E(X^{2})-E(X^{2})}\sqrt{E(Y^{2})-E(Y^{2})}}$$
where *X* represents the original signal and *Y* represents each mode. The absolute value of $\rho_{X,Y}$ is used to determine the correlation of each mode with the original signal. $\left|\rho_{X,Y}\right|=1$ means there is a complete cross-correlation, and $\left|\rho_{X,Y}\right|=0$ means it is completely irrelevant.

When the SNR of the detection signal is relatively high, the noise interference in the signal is less, and the mode with a high correlation coefficient can reflect more fault information. When there is a signal with a low SNR, the noise interference in the signal is greater. The mode with a high ASN takes both the pulse characteristics and cyclostationarity characteristics into account to reflect more fault information. Therefore, it is reasonable to use the correlation coefficient as the adaptive factor to correct the value. The product of the two can play a complementary role with the modified adaptive average spectral negentropy (AASN) shown in the following formula:
(16)$$\Delta I_{c}(IMF_{i})=\rho_{X,Y_{i}}\times \Delta I_{1/2}(IMF_{i})$$


### 3.3. Fault Feature Extraction Based on the AEWT

When the *i*-th mode corresponding to the highest value in the AASN is selected as the center of fault information, there may be a case where the demodulation band is relatively narrow, so it is necessary to consider whether to merge adjacent modes. In this paper, the correlation coefficient is used as the index to judge the merger. According to practical experience, if the adjacent mode is greater than 0.25, it should be merged with the central mode [32]. The framework of the proposed method is illustrated in Figure 1.

For the detection of periodic fault features, the following steps are given to apply an AASN to the EWT (AEWT) and construct an optimal resonance frequency band in a completely data-driven way:
Step 1:Fourier spectrum *X*(*f*) is obtained by performing the DFT on the input signal *x*(*t*).Step 2:Apply a parameterless clustering scale-space histogram segmentation boundary to *X*(*f*).Step 3:Application EWT is used to decompose the input signal into a set of empirical modes.Step 4:Calculate the ASN and correlation coefficient values on each mode.Step 5:The AASN is calculated according to Equation (15). The mode corresponding to the highest value is selected as the center band.Step 6:Determine whether the correlation coefficient $\rho_{X,Y}$ of the adjacent mode is greater than 0.25, and if the requirement is met, the reconstruction signal is composed.Step 7:Fault types are diagnosed by observing Hilbert envelope spectrum.


## 4. Simulations and Comparative Analysis

A simulation experiment is made to compare the effectiveness of the proposed method with other methods. The simulation signal is composed of a periodic impulse component and strong background noise to simulate the vibration signal with impulse characteristics when defects occur in rotating machinery:
(17)$$x(t)=y_{0}e^{-2\pi f_{n}\xi t}\times \sin(2\pi f_{n}\sqrt{1-\xi^{2}t})$$
where the displacement constant of the bearing is *y*_0_ = 2.5, the relative damping coefficient is ξ=0.1, the resonance frequency *f_n_* = 900 Hz, *t* is the sampling moment, the impact fault period is *T* = 0.00625 s (the characteristic frequency is *f_c_* = 160 Hz), the sampling frequency is *f_s_* = 12 kHz and simulated data length is 4096. The periodic vibration pulse signal is generated as shown in Figure 2a. Meanwhile, adding SNR = −4 dB white Gaussian noise to the original signal, the waveform of the mixed signal is illustrated in Figure 2b. 

It can be seen that due to the influence of noise interference, it is difficult to find pulses in the time domain. To test the effectiveness of the proposed method, the AEWT is applied to analyze the analog signal.

The parameterless clustering scale-space histogram method is applied to the detected signal, and the boundaries of each mode are shown in Figure 3. The ASN of all modes, the correlation coefficient with the original signal, and the results of the AASN are provided in Table 1. In these modes, the highest value of the AASN appears in mode 3. Among them, the highest value of the ASN appears in mode 3, and the highest correlation coefficient appears in the mode 15, so the AASN is consistent with the results of the ASN.

For simplicity, only modes 1–6 near the determination conditions are shown in Figure 4a, as they may best meet the requirements of the characteristic resonance band. The corresponding envelope spectrum is presented in Figure 4b. Due to the influence of a strong background noise, the parameterless scale-space segments the frequency band too narrowly. It can be seen that in these modes, fault-related modulation information is not identified. As shown in the dotted circle, the envelope spectrum of mode 3 shows no feature information. Therefore, we should consider merging the adjacent modes to construct the optimal resonance demodulation frequency band. According to the correlation coefficient merging strategy, it is determined whether the mode adjacent to the central mode satisfies the merge condition. As shown in Table 1, IMF1, IMF2, and IMF4 meet the requirements. 

The envelope spectrum of the combined mode is shown in Figure 5, which can clearly identify the modulation characteristics of the fault correlation frequency and multiple harmonic waves. Under the influence of a strong noise, due to the large number of interference components, there is a large deviation when using a correlation analysis as the basis for the merger. However, the ASN can reflect the real results more accurately, and the product can play the role of an adaptive correction. The simulation results show that the proposed fault center mode selection principle and the correlation coefficient combination principle are both reliable and effective.

To verify the superiority of the proposed method, we also use the EMD method for the test signal. The EMD decomposition results are presented in Figure 6a. For simplicity, we only show the top five IMFs. It can be seen that the EMD is sensitive to noise, and no periodic pulse information can be seen in the decomposed IMF. The corresponding envelope spectrum is shown in Figure 6b. Although the fault frequency can be extracted by IMF1 and IMF2, it is not obvious and the relevant harmonic components cannot be extracted.

At the same time, in order to verify the superiority of the proposed method in selecting the center position of the resonant frequency band, the spectral kurtosis method is used for the demodulation analysis of the test signal, and after three Kurtogram layers, the signal is decomposed into several sub-signals of the same bandwidth, as shown in Figure 7a. The highest Kurtosis band is shown in the dotted circle, with a center frequency of 4250 Hz and a bandwidth of 1500 Hz. Its envelope spectrum is displayed in Figure 7b. It can be observed that the fault feature information cannot be extracted in the envelope spectrum because of the large error in selecting the center position.

In the actual industrial environment, noise interference is unavoidable. The parameterless scale-space histogram method is sensitive to noise, which leads to the narrow boundary division of the spectrum. It is difficult to demodulate the fault information in corresponding modes. Therefore, the reconstructed mode can be effectively used for fault feature recognition by selecting the center of the resonance band and reasonably combining the adjacent modes. The result analysis in the analog signal shows that the AEWT has a better performance than the EMD and FK methods. To further verify the effectiveness of this method, it is applied to the task of actual mechanical fault identification.

## 5. Application to Roller Bearings Testing

In this section, the proposed method is applied to analyze the outer ring fault signal of the Case Western Reserve University (CWRU) bearing data set [33] and the National Aeronautics and Space Administration (NASA) bearing data set [34], respectively. It is also compared with the FK, EMD and the modified empirical wavelet transform (MEWT) method proposed in [17]. The bearing with normal and degraded bearing [35] is shown in Figure 8.

### 5.1. Case 1: Analysis of CWRU Outer Ring Fault Vibration Signal

As shown in Figure 9, the detailed bearing parameters are as follows: the bearing model is 6205-2RSJEMSKF, and three acceleration sensors are installed at the motor drive end at the 3 o’clock, 6 o’clock and 12 o’clock positions of the bearing housing (in this paper, the vibration signal of the bearing outer ring at 6 o’clock is selected for the experimental analysis). The motor speed is 1797 rpm (rotational frequency is 1797/60 Hz = 29.95 Hz), the sampling frequency is 12 kHz, and the experimental data length is 2048.

The outer ring vibration signal with a damage diameter of 0.007 inches (0.01778 cm) is used for experimental analysis. According to the calculation formula in [36], the characteristic frequency of bearing outer ring fault can be obtained:
(18)$$\mathrm{BPFO}=\frac{1}{2}f_{s}(1-\frac{d_{r}}{D_{m}}\cos\theta )n_{r}$$


Among them, the diameter of the roller element *D_m_* = 39 mm, the pitch diameter *d_r_* = 7.938 mm, the number of roller elements *n_r_* = 1 and the contact angle *θ* = 0°. Therefore BPFO = 107.3648 Hz. The time domain waveform of the outer ring fault vibration original signal is shown in Figure 10. It can be seen that the noise interference in the signal is weak and the defective pulse characteristic is obvious. The proposed method is applied to the abnormal condition of the outer ring of rolling bearing in CWRU experimental platform to explore the effectiveness of the proposed method in a relatively pure environment.

The Fourier spectrum of the outer ring fault signal is subjected to parameterless scale-space segmentation, and a total of nine modes are decomposed, as shown in Figure 11. The ASN of all modes, the correlation coefficient with the original signal, and the results of the AASN are given in Table 2. In these modes, the highest value of the AASN appears in mode 7. Among them, the highest value of ASN appears in mode 1, and the highest correlation coefficient appears in the model 7, so the decision result of the AASN is consistent with the decision result of a high correlation.

The envelope spectrum corresponding to mode 7 is illustrated in Figure 12, which can clearly identify the fault frequency and 2-5 doubling frequency information. It is proven that the mode with a strong correlation can better reflect the fault pulse information in a relatively pure environment, and the center position of the AASN selection is accurate. To find the best resonant demodulation frequency band, it is necessary to examine whether or not to combine adjacent modes. As shown in Table 2, mode 6 meets the merger requirements. The combined envelope spectrum is shown in Figure 13. It is noted that both the fault frequency kurtosis and frequency doubling information are increased.

To verify the superiority of the proposed method, the same signal is also compared and analyzed through the EMD. The envelope spectrum in Figure 14 shows the most abundant mode of the fault characteristics in the decomposition results. It can be seen that both the kurtosis of the failure frequency and the number of harmonics are weakened due to modal mixing and other phenomena in the EMD. At the same time, in order to verify the rationality of the proposed method in reconstructing the resonant frequency band, the test signal is demodulation and analyzed by the spectral kurtosis method. After the three-layer kurtogram, the frequency band with the highest kurtosis is selected, with a center frequency of 3000 Hz, and a bandwidth of 2000 Hz. As shown in the dotted circle in Figure 15. By comparing this with Figure 10, it is determined that the center position and frequency bandwidth selected by SK is similar to the proposed method. This finding shows that both methods can locate the optimal filter band when the impulse interference in the system is less.

### 5.2. Case 2: Analysis of NASA Outer Ring Fault Vibration Signal

The effectiveness of the proposed method is evaluated by using bearing vibration signal data collected by NASA. The experimental platform is shown in Figure 16. Double-row Rexnord ZA-2115 roller bearings are mounted on the same spindle. PCB353B33 acceleration sensors are fitted to each bearing shaft. The rotation speed of the spindle is constant at 2000 rpm, i.e., the rotation frequency is 2000/60 Hz = 33.3 Hz. Each experiment has 20,480 data points and the sampling frequency of the acquisition system is 20 kHz. The outer ring signal in the experiment is at bearing 1 of the NO.2 dataset. In this paper, the outer ring fault vibration signal obtained at the end of the experimental failure is selected to verify the proposed method. The length of the experimental data is 2048. Among them, the diameter of the roller element *D_m_* = 71.501 mm, the pitch diameter *d_r_* = 8.407 mm, the number of roller elements *n_r_* = 1 and the contact angle *θ* = 15.17°. BPFO = 236.4 Hz, which is computed according to Equation (18).

The original fault vibration signal waveform of the outer ring is shown in Figure 17a. It can be seen that the NASA dataset is affected by weak noise, and the periodic pulse characteristic cannot be clearly seen in the time domain diagram. At the same time, in order to verify the effectiveness of the proposed method under the variation of noise intensity, Gaussian white noise with SNR = 10 dB is added to the original signal, and the time domain waveform of the mixed signal is shown in Figure 17b.

#### 5.2.1. Original Fault Vibration Signal Analysis

Next, we apply the proposed method to the original fault signal of the outer ring to verify the performance of the actual mechanical fault vibration signal when it is contaminated by weak noise. The parameterless scale-space segmentation of the Fourier spectrum of the outer ring fault signal is as shown in Figure 18, and a total of 14 modes are decomposed. Table 3 shows the ASN of all modes, the correlation coefficients with the original signal and the results of the AASN. In these modes, the highest value of the AASN appears in mode 2, the highest value of ASN appears in mode 3, and the highest value of correlation coefficient appears in mode 2, so the results of AASN are consistent with those of high correlation. Figure 19 shows six adjacent modes that meet the criteria, and the envelope spectrum corresponding to mode 2 is indicated in Figure 20. As can be seen, although the frequency 234.4 Hz, which is close to the fault frequency (236.4 Hz), and the harmonic component 58.59 Hz can be extracted, due to the excessive decomposition of signals, the fault frequency is weak and fewer harmonic components are extracted. Therefore, it is necessary to consider merging the adjacent modes to construct the optimal demodulation frequency band. As shown in Table 3, modes 1, 3 and 4 meet the merger requirements. As shown in Figure 21, the combined envelope spectrum fault characteristic frequency is more prominent, and the multiple harmonic components can be clearly identified.

As a comparison, the EMD and FK methods are used to analyze the same signals to verify the superiority of the proposed method in the case of weak noise. By comparing and observing each IMF, we choose the mode with the most abundant fault information for the envelope spectrum analysis. As shown in Figure 22, it can be observed that although the EMD shows a better decomposition ability in a weaker noise environment, the characteristic frequency and the harmonic component amplitude it extracts are all weaker than the proposed method. Similarly, as shown by the dotted circle in Figure 23, the center of the demodulation band displayed by the kurtogram is 1667 Hz, and the bandwidth is 3333 Hz. It is determined that the position and size of the filter are quite similar to the proposed method, which shows that when the central frequency can be precisely determined, the merging criterion utilized in the construction of the filter size is effective and reasonable.

#### 5.2.2. Analysis of Noise-Adding Fault Vibration Signal

To explore the robustness of the proposed algorithm under the background of strong noise, Gaussian white noise with SNR = 10 dB is added to the original signal. The proposed method is applied to the mixed signal, and the MEWT method is applied to the same vibration signal for comparative analysis. As shown in Figure 24a, the parameterless clustering scale-space segmentation is performed on the Fourier spectrum of the mixed signal, and a total of 15 empirical modes are decomposed; meanwhile, according to the suggestion in [18], the scale parameter is set to $\sqrt{n}=3fc$. Results of the MEWT method are shown in Figure 24b for a total of four empirical modes decomposed. 

As the interference component in the signal increases, the parameterless clustering method becomes too narrow for spectrum segmentation. Therefore, it is necessary to comprehensively consider the characteristic information carried on each frequency band and reconstruct the appropriate resonance demodulation bandwidth. As shown in Table 4, the highest value of the AASN is the same as the highest value of the ASN, which appears in mode 2, and the highest value of correlation coefficient appears in mode 13. Obviously, compared with the analysis results of the original signal in the previous section, there is a misjudgment in the selection of mode 13. It can also be concluded that when the background noise is enhanced, the mode with a strong correlation may not reflect the pulse characteristics of the signal to a greater degree. To further analyze the feature information of each mode, modes 1–6 near the determination condition are shown in Figure 25a, and the corresponding envelope spectra are shown in Figure 25b. As shown in the dotted circle, the fault frequency can be found in mode 2, which indicates that the center position selected by AASN is accurate. However, due to the narrow frequency band setting, the characteristics are very weak and no more harmonic components can be found. Therefore, it is necessary to consider combining adjacent modes to set up the optimal resonant demodulation bandwidth. According to the results of Table 4, mode 1, mode 3 and mode 4 meet the merging requirements. The combined envelope spectrum is shown in Figure 26. Although there are more interference components, the reconstructed envelope spectrum can clearly identify the fault frequency and harmonic components. Meanwhile, as a comparison, Figure 27 shows the envelope spectrum of mode 1 in Figure 24b, and it can be found that the burrs in the envelope spectrum are increased and the fault characteristics and harmonic component peaks are weakened. This means that the visualization ability of the proposed method is better than that of the MEWT method under different noise intensities.

To demonstrate the superiority of the proposed method, the same data are applied to the EMD and FK method. Figure 28a shows the first six IMFs of the EMD and the corresponding envelope spectra. It can be seen from Figure 28b that due to the influence of noise, only the weak fault frequency can be identified in the envelope spectrum of mode 1, as shown in the dotted line ellipse, while the fault frequencies and harmonic components cannot be identified in other envelope spectra. Similarly, as shown by the dotted circle in Figure 29a, the center of the demodulation band displayed by the kurtogram is 8125 Hz, and the bandwidth is 1250 Hz, and the selected center position is larger than that of the proposed method. Figure 29b shows the corresponding envelope spectrum. Since the resonance demodulation band is not determined, the fault frequency and harmonic components cannot be found in the envelope spectrum. It is demonstrated that when the background noise in the system is strong, the center position selected by the proposed method is more accurate and effective than the center position selected by the kurtogram.

## 6. Discussion and Conclusions

Although the simulation results and actual data analysis have proved the effectiveness of this method, there is still some useful information worth further mining.

### 6.1. Discussion

(1) Under actual operating conditions, the mechanical vibration signals will inevitably be exposed to noise pollution. Based on [18], this paper analyzes the influence of selecting fixed scale parameter on the feature extraction ability when the background noise intensity changes. At the same time, the strong noise increases the difficulty of feature extraction method design and greatly affects the uncertainty of the experimental results. Therefore, the noise preprocessing mechanism will play a positive role in the use of the method.

(2) The randomness of the noise leads to many narrow-band modes in the decomposition result. When the center frequency is selected, the fusion of the correlation coefficient and ASN is used as the judgment index, and good results are obtained. However, the quantitative influences of the squared envelope, square envelope spectrum and correlation coefficient on the selection results when the background noise intensity changes have not been analyzed. Therefore, the selection of each parameter should be further optimized to improve the effectiveness of this method.

(3) Compared with the traditional binary frequency band segmentation method, the scale-space histogram segmentation method adopted in this paper has more physical significance. In the mode adjacent to the center frequency, the mode whose correlation coefficient is greater than 0.25 is combined to construct the optimal resonance demodulation frequency band. In other words, adjacent modes with default correlation coefficients less than 0.25 are noise, to achieve the purpose of noise elimination. In actual engineering, this parameter deserves further optimization.

### 6.2. Conclusions

(1) The simulation signals of strong background noise, the relatively pure CWRU data set of the experimental environment, the NASA dataset with weak noise and the NASA dataset for SNR=10 dB Gaussian white noise were selected for subsequent analysis. Under background noises of different intensities, the fault characteristic frequencies and harmonic characteristics extracted by the proposed method are more effective and obvious. In contrast, MEWT, FK, EMD and other methods are sensitive to noise, and the extraction effect is feeble or even invalid.

(2) In a practical environment, due to the influence of background noise, parameterless clustering scale-space histogram will divide the spectrum into a variety of narrow bands. When an ASN is used to select the center frequency, the high value will appear in numerous bands or even a misjudgment. In this paper, the correlation coefficient is used as an adaptive factor to modify the ASN, and the information fusion between them improves the selection accuracy. Through the experimental verification of different intensity noise backgrounds, the AASN can effectively locate the fault center frequency and show better universality and robustness.

(3) A large number of experiments show that when the frequency band is set too narrow, the fault information will be weakened or even unable to be extracted. When the frequency band is too wide, more serious noise interference will be introduced, and the feature information will be submerged by the noise signal. In this paper, the correlation coefficient is selected as the judgment index of frequency band merging. If the correlation coefficient of the adjacent mode is greater than 0.25, it is merged with the central mode. In other words, the correlation coefficient is equivalent to the adaptive selection of the frequency band according to the noise intensity; therefore, the feasibility of band merging is verified in all experiments. In our future work, we will consider more reasonable approaches to optimize the parameters of the proposed method and add other denoising techniques.

## Figures and Tables

**Figure 1 entropy-21-00135-f001:**
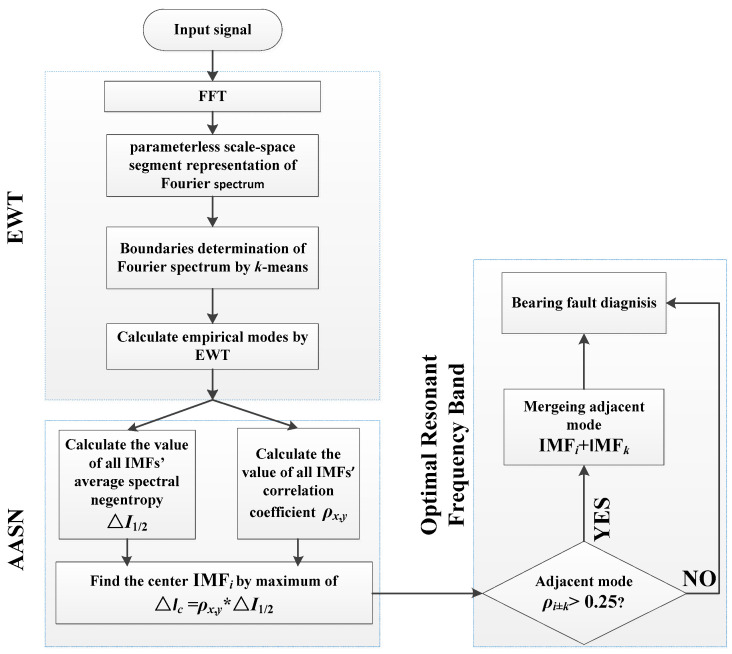
Flow chart of the overall procedure of AEWT.

**Figure 2 entropy-21-00135-f002:**
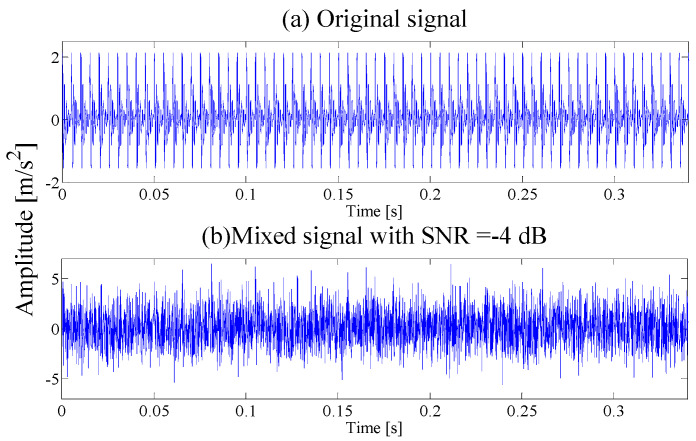
(**a**) Time-domain Waveform of Original signal; (**b**) Time-domain Waveform of SNR = −4 dB mixing Signal.

**Figure 3 entropy-21-00135-f003:**
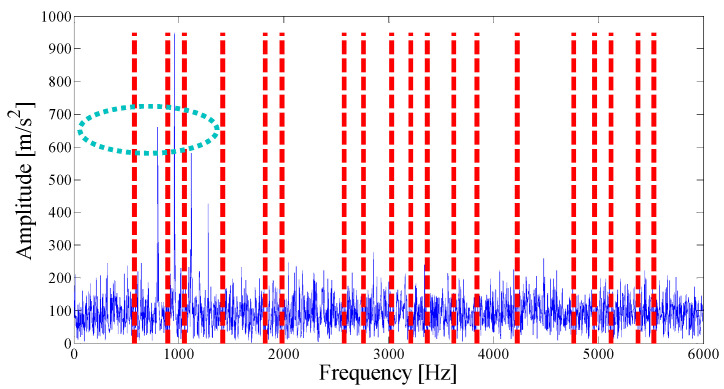
Analyzed result of the boundaries detected using the parameterless scale-space method.

**Figure 4 entropy-21-00135-f004:**
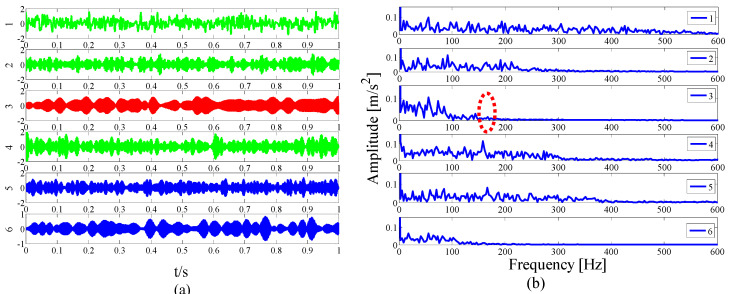
Analysis results by the EWT method of (**a**) the empirical modes 1–6 and (**b**) The Hilbert envelope spectrums.

**Figure 5 entropy-21-00135-f005:**
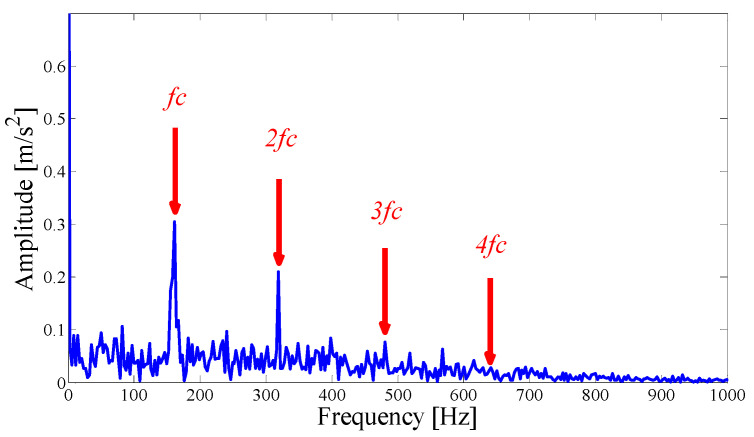
Hilbert envelope spectrum of the optimal merging mode.

**Figure 6 entropy-21-00135-f006:**
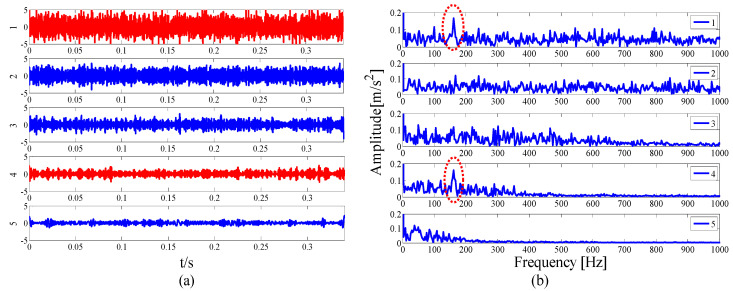
Analysis results by the EMD method of (**a**) the waveforms first five IMFs and (**b**) The Hilbert envelope spectra.

**Figure 7 entropy-21-00135-f007:**
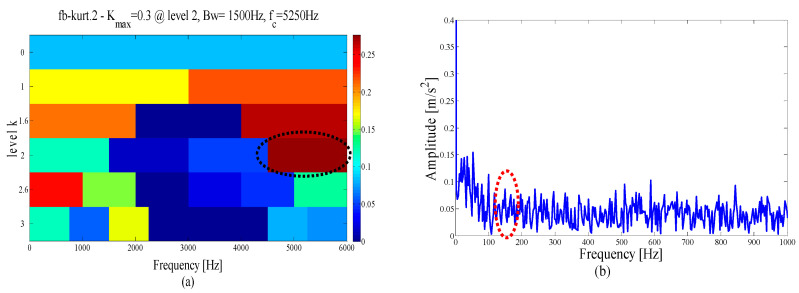
Results using spectral kurtosis: (**a**) kurtogram and (**b**) envelope spectrum.

**Figure 8 entropy-21-00135-f008:**
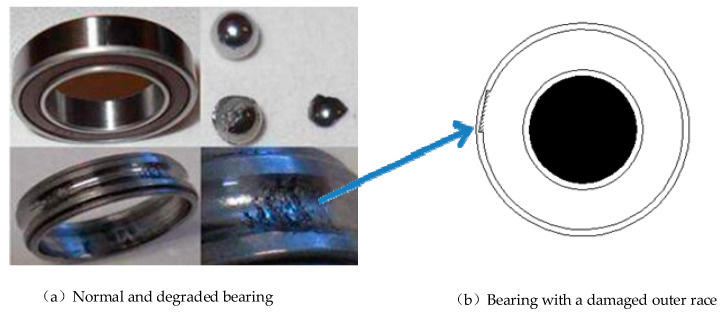
Illustration of the normal and degraded bearings.

**Figure 9 entropy-21-00135-f009:**
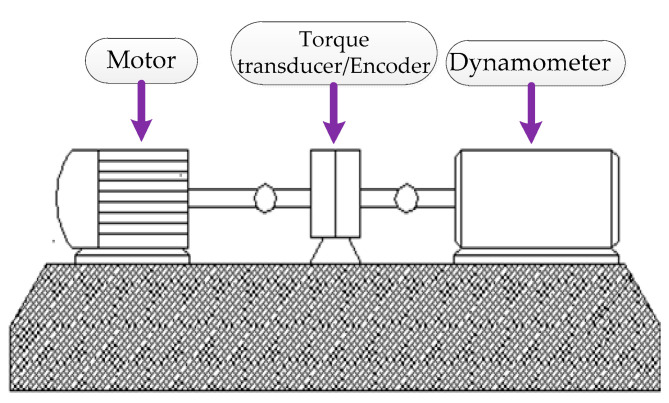
Illustration of the bearing experiment platform.

**Figure 10 entropy-21-00135-f010:**
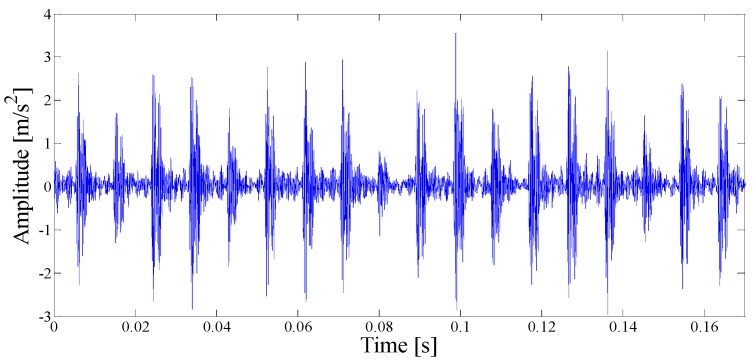
Time-domain waveform of the original signal.

**Figure 11 entropy-21-00135-f011:**
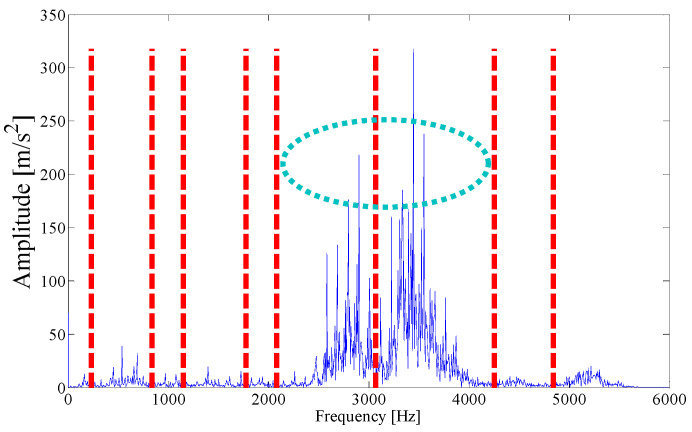
Analyzed result of the boundaries detected using parameterless scale-space method.

**Figure 12 entropy-21-00135-f012:**
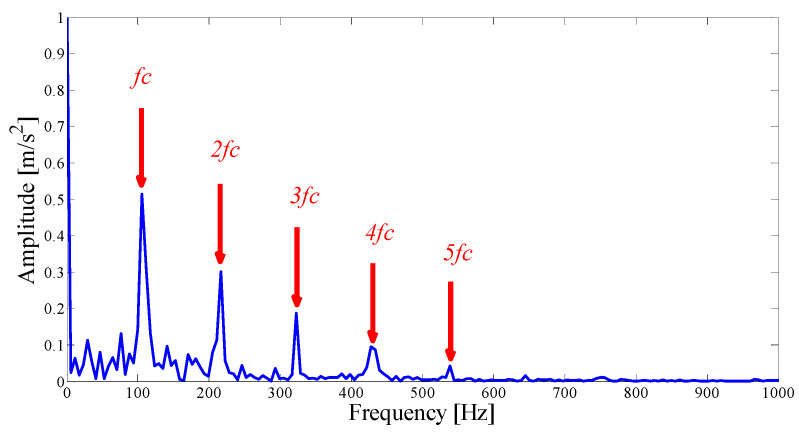
The Hilbert envelope spectrum of IMF7.

**Figure 13 entropy-21-00135-f013:**
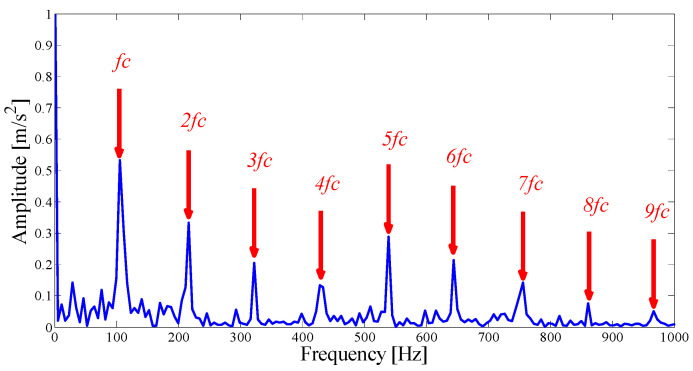
The Hilbert envelope spectrum of the optimal merging mode.

**Figure 14 entropy-21-00135-f014:**
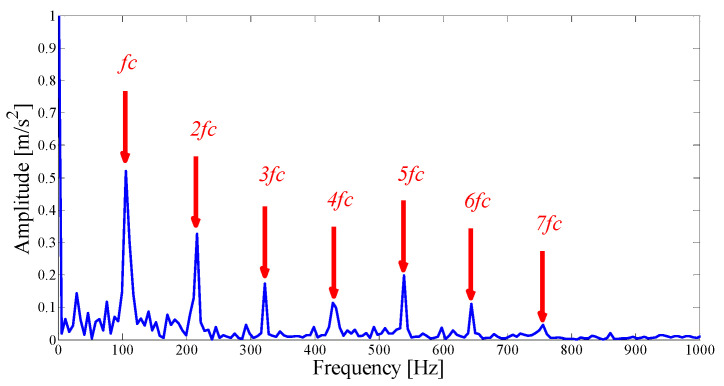
The Hilbert envelope spectrum analysis results by the EMD method.

**Figure 15 entropy-21-00135-f015:**
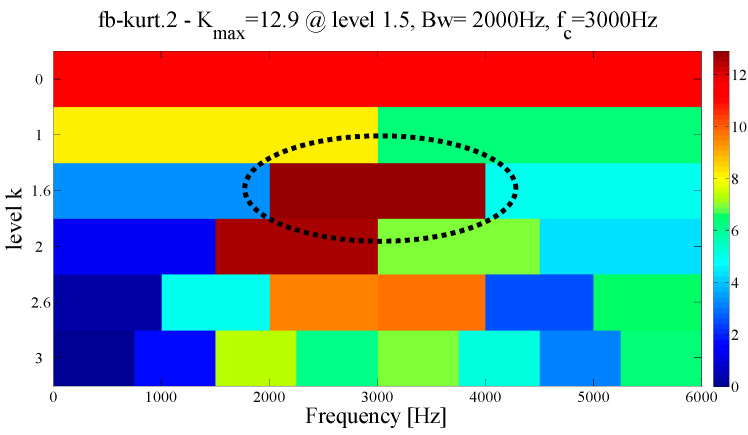
Results using spectral kurtosis.

**Figure 16 entropy-21-00135-f016:**
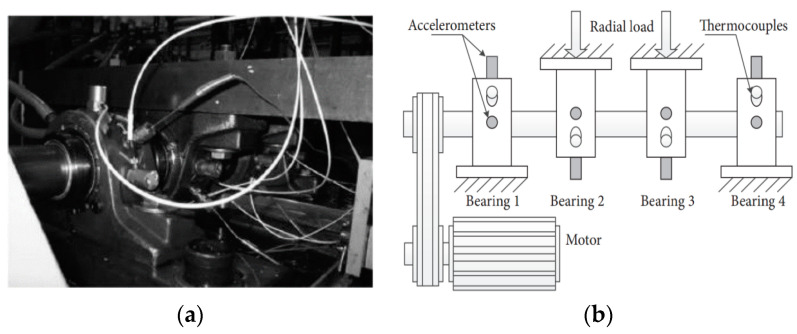
Illustration of the bearing experiment platform. (**a**) Bearing test rig; (**b**) Sensor placement illustration.

**Figure 17 entropy-21-00135-f017:**
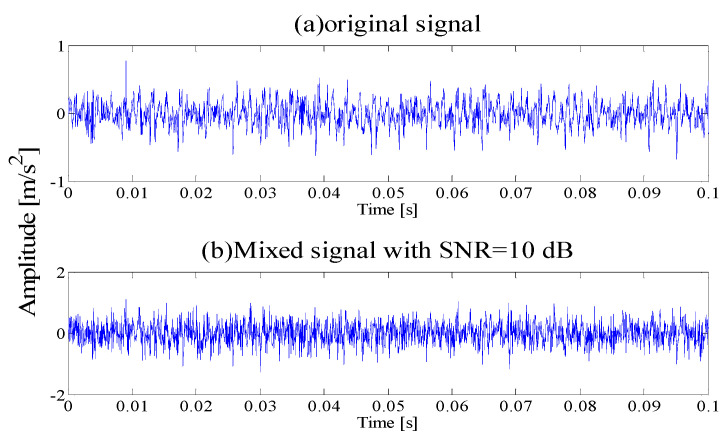
(**a**) Time-domain waveform of the original signal; (**b**) Time-domain waveform of SNR = 10 dB mixing signal.

**Figure 18 entropy-21-00135-f018:**
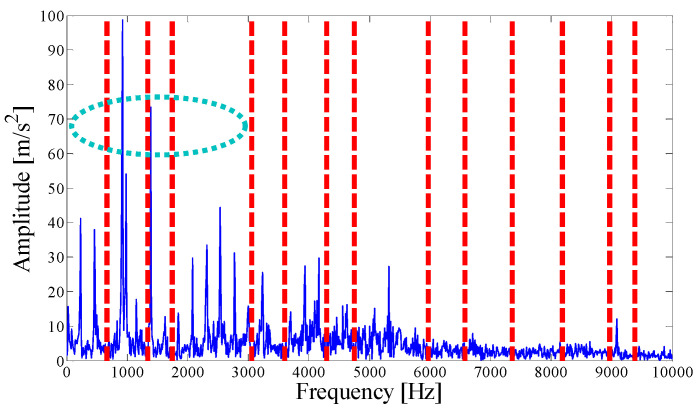
Analyzed result of the boundaries detected using the parameterless scale-space method.

**Figure 19 entropy-21-00135-f019:**
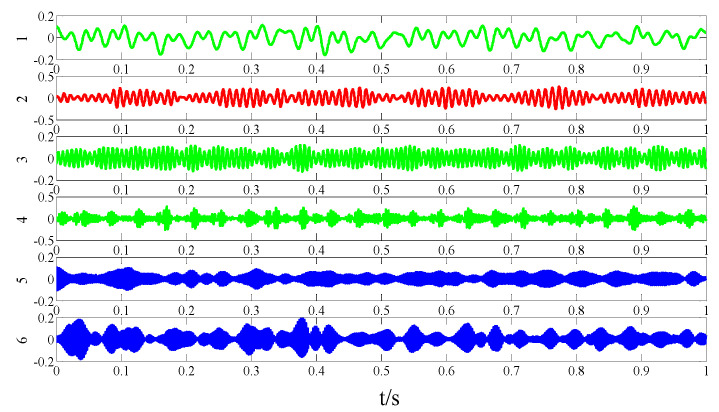
Analysis results by the EWT method of the empirical modes 1–6.

**Figure 20 entropy-21-00135-f020:**
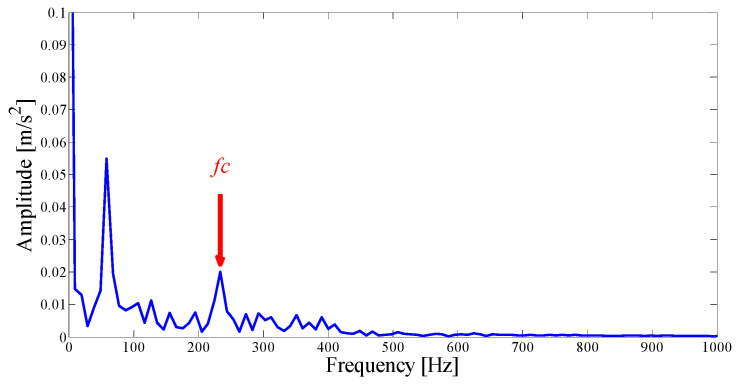
The Hilbert envelope spectrum of IMF2.

**Figure 21 entropy-21-00135-f021:**
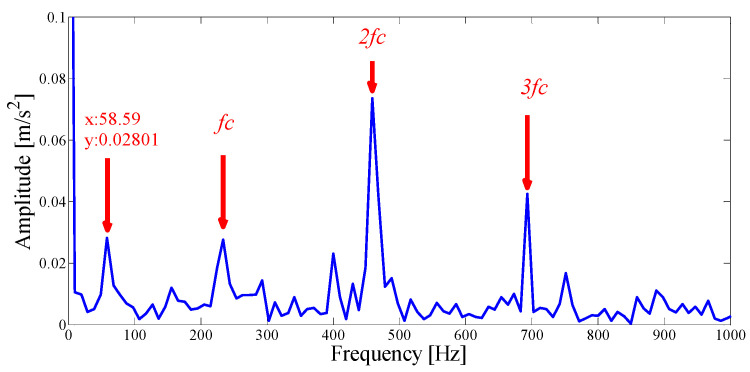
Hilbert envelope spectrum of the optimal merging mode.

**Figure 22 entropy-21-00135-f022:**
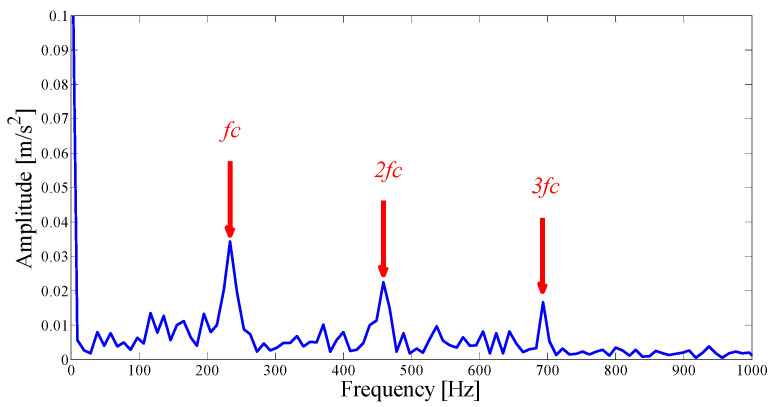
The Hilbert envelope spectrum analysis results by the EMD method.

**Figure 23 entropy-21-00135-f023:**
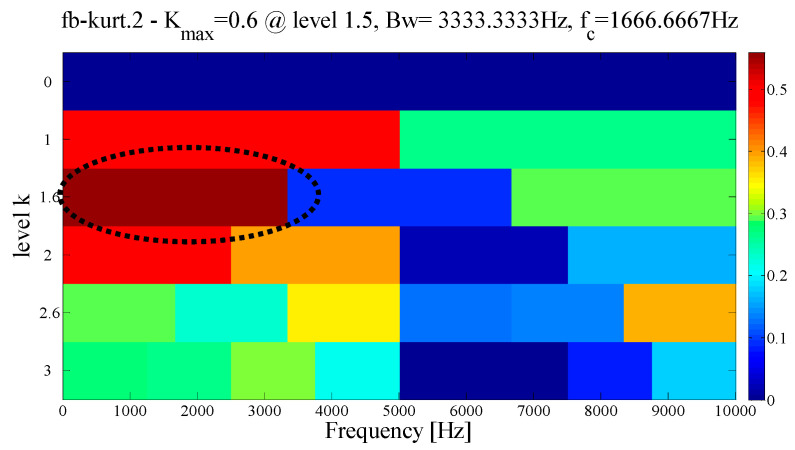
Results using spectral kurtosis.

**Figure 24 entropy-21-00135-f024:**
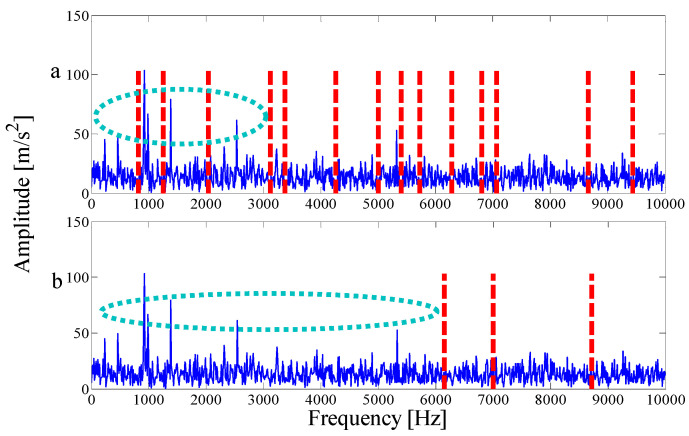
The analyzed result of the boundaries detected: (**a**) The parameterless scale-space method. (**b**) The MEWT method.

**Figure 25 entropy-21-00135-f025:**
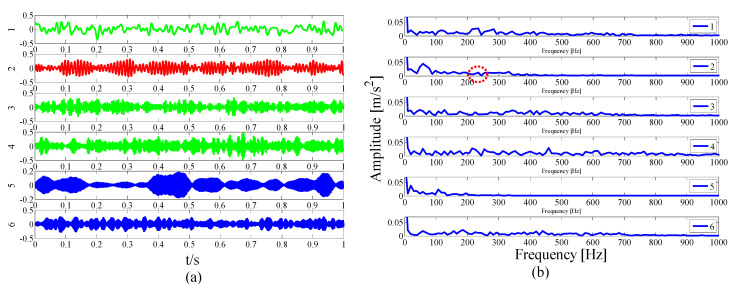
Analysis results by the EWT method of (**a**) the empirical modes 1-6 and (**b**) The envelope spectra.

**Figure 26 entropy-21-00135-f026:**
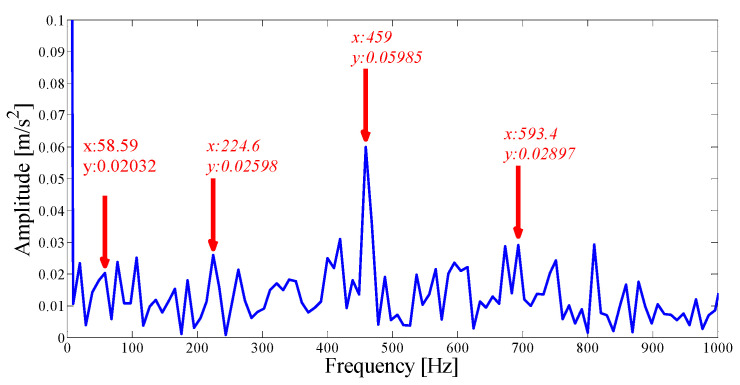
Hilbert envelope spectrum of the optimal merging mode.

**Figure 27 entropy-21-00135-f027:**
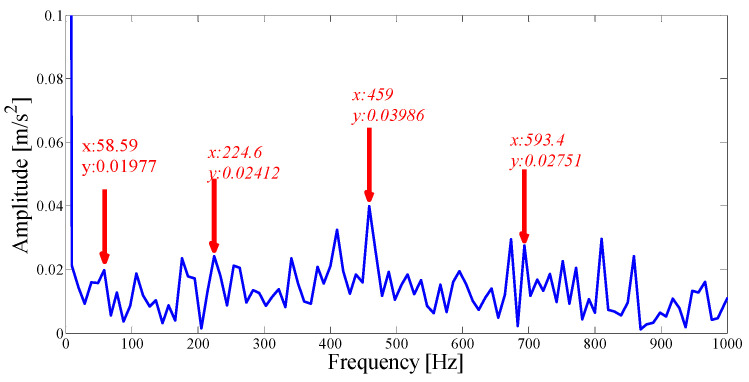
The envelope spectrum analysis by MEWT.

**Figure 28 entropy-21-00135-f028:**
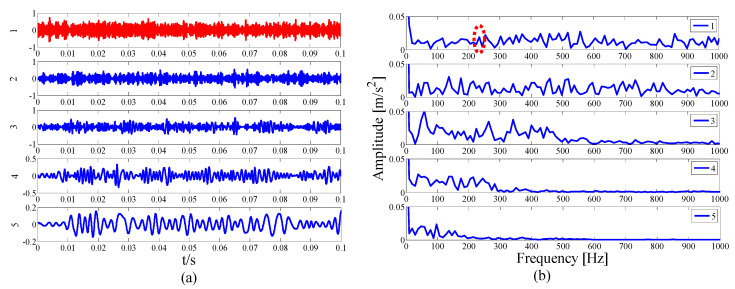
Analysis results by the EMD method of: (**a**) the waveforms first five IMFs and (**b**) The Hilbert envelope spectrums.

**Figure 29 entropy-21-00135-f029:**
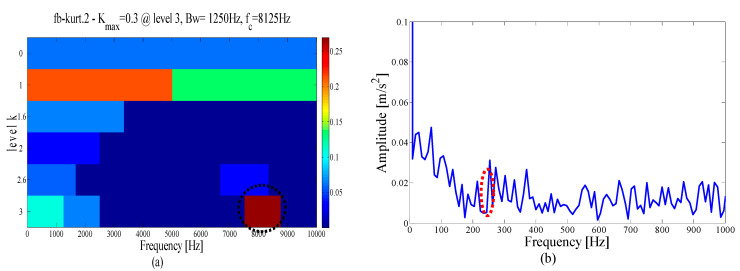
Results using spectral kurtosis: (**a**) kurtogram and (**b**) Hilbert envelope spectrum.

**Table 1 entropy-21-00135-t001:** The value of the Average Spectral Negentropy ∆*I*_1/2_, the Correlation coefficient *ρ*_*X*,*Y*_ and the Adaptive Average Spectral Negentropy ∆*I_C_*.

*IMF*	∆*I*_1/2_	*ρ* _*X*,*Y*_	∆*I_C_*	*IMF*	∆*I*_1/2_	*ρ* _*X*,*Y*_	∆*I_C_*
1	5.7755	**0.2936**	1.6954	11	6.4204	0.1561	1.0022
2	7.3191	**0.2682**	1.8896	12	6.1135	0.1918	1.1728
3	**7.6186**	**0.2883**	**2.1966**	13	6.2320	0.1742	1.0858
4	6.7521	**0.2960**	1.9989	14	5.9637	0.2477	1.4773
5	5.9245	0.2467	1.4795	15	5.8851	**0.3126**	1.8399
6	6.3692	0.1491	0.9499	16	6.3084	0.1737	1.0958
7	5.6528	0.3050	1.7238	17	6.4203	0.1740	1.1174
8	6.3221	0.1700	1.0747	18	6.3126	0.2104	1.3284
9	6.2089	0.2121	1.3168	19	6.4710	0.1717	1.1113
10	6.6168	0.1734	1.1472	20	6.0843	0.2624	1.5964

**Table 2 entropy-21-00135-t002:** The value of the Average Spectral Negentropy ∆*I*_1/2_, the Correlation coefficient *ρ*_*X*,*Y*_ and the Adaptive Average Spectral Negentropy ∆*I_C_*.

*IMF*	∆*I*_1/2_	*ρ* _*X*,*Y*_	∆*I_C_*	*IMF*	∆*I*_1/2_	*ρ* _*X*,*Y*_	∆*I_C_*
1	**7.6090**	0.0204	0.1551	6	6.2622	**0.5325**	3.3348
2	6.3599	0.0788	0.5012	7	6.1155	**0.8502**	**5.1992**
3	6.3114	0.0284	0.1789	8	5.7584	0.0442	0.2548
4	6.2948	0.0444	0.2797	9	5.7374	0.0814	0.1668
5	6.1204	0.0278	0.1702				

**Table 3 entropy-21-00135-t003:** The value of the Average Spectral Negentropy ∆*I*_1/2_, the Correlation coefficient *ρ*_*X*,*Y*_ and the Adaptive Average Spectral Negentropy ∆*I_C_*.

*IMF*	∆*I*_1/2_	*ρ* _*X*,*Y*_	∆*I_C_*	*IMF*	∆*I*_1/2_	*ρ* _*X*,*Y*_	∆*I_C_*
1	6.4651	**0.3079**	1.9903	8	6.0680	0.2747	1.6668
2	6.7327	**0.5824**	**3.9212**	9	5.6489	0.1038	0.5865
3	**7.1859**	**0.3234**	2.3240	10	5.6357	0.1090	0.6140
4	6.3065	**0.4246**	2.6780	11	5.4719	0.0934	0.5111
5	6.4987	0.2032	1.3207	12	5.6168	0.1023	0.5745
6	6.0117	0.3014	1.8122	13	6.8529	0.0778	0.5329
7	5.9508	0.1887	1.1230	14	5.6431	0.0555	0.3131

**Table 4 entropy-21-00135-t004:** The value of the Average Spectral Negentropy ∆*I*_1/2_, the Correlation coefficient *ρ*_*X*,*Y*_ and the Adaptive Average Spectral Negentropy ∆*I_C_*.

*IMF*	∆*I*_1/2_	*ρ* _*X*,*Y*_	∆*I_C_*	*IMF*	∆*I*_1/2_	*ρ* _*X*,*Y*_	∆*I_C_*
1	5.8326	**0.2946**	1.7182	9	5.9337	0.1796	1.0656
2	**6.5265**	**0.3609**	**2.3557**	10	5.6078	0.2080	1.1664
3	6.4852	**0.3098**	2.0091	11	5.6867	0.2146	1.2206
4	5.7762	**0.3534**	2.0411	12	5.9589	0.1649	0.9827
5	6.3258	0.1686	1.0668	13	5.2600	**0.3758**	1.9768
6	5.4575	0.2721	1.4849	14	5.5263	0.2492	1.3771
7	5.4696	0.2451	1.3407	15	5.8034	0.2059	1.1947
8	6.3494	0.2085	1.3238

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
