# Peer review of "An Optimal Resonant Frequency Band Feature Extraction Method Based on Empirical Wavelet Transform"

_entropy, 2019, doi:10.3390/e21020135_

Round 1
Reviewer 1 Report
1) Figures should have better quality
2) Why the topic is essential. Please add image of application.
Please add photo of experimental setup.
The paper is boring now.
Figure 8 should be replaced. I saw this photo 10 times. It is not original.
please make another photo.
3) Please add easy block diagram/ step by step what are you analysed, what parameters/values
4) Please add 2-3 sentences about future analysis.
Please check English.
5) Too many references to "Mech. Syst. Signal Process".
You can cite one journal 1-4 times in one paper, not more.
Don't sell citations. It is analysed.
6) The authors should cite new references (2016-2019 Web of Science).
for example 20-30 new references.
Please show that you have new knowledge about diagnostic methods.
for example.
Fault diagnosis of single-phase induction motor based on acoustic signals
By:Glowacz, A (Glowacz, Adam)[ 1 ]
MECHANICAL SYSTEMS AND SIGNAL PROCESSING
Volume: 117 Pages: 65-80
DOI: 10.1016/j.ymssp.2018.07.044
Published:FEB 15 2019
Multi-dimensional variational mode decomposition for bearing-crack detection in wind turbines with large driving-speed variations
By:Li, ZX (Li, Zhixiong)[ 1,3,4 ] ; Jiang, Y (Jiang, Yu)[ 3,4,5 ] ; Guo, Q (Guo, Qiang)[ 6 ] ; Hu, C (Hu, Chao)[ 1,2 ] ; Peng, ZX (Peng, Zhongxiao)[ 5 ]
RENEWABLE ENERGY
Volume: 116 Pages: 55-73 Part: B
DOI: 10.1016/j.renene.2016.12.013
Published:FEB 2018
Optimised ensemble empirical mode decomposition with optimised noise parameters and its application to rolling element bearing fault diagnosis
By:Zhang, C (Zhang, Chao)[ 1,2 ] ; Li, ZX (Li, Zhixiong)[ 2,3,4 ] ; Chen, S (Chen, Shuai)[ 1 ] ; Wang, JG (Wang, Jianguo)[ 1 ] ; Zhang, XG (Zhang, Xiaogang)[ 2 ]
INSIGHT
Volume: 58 Issue: 9 Pages: 494-501
DOI: 10.1784/insi.2016.58.9.494
Published:SEP 2016
Glowacz A.: Recognition of acoustic signals of commutator motors, APPLIED SCIENCES, 8 (12), 2630, 2018. https://doi.org/10.3390/app8122630
Integrated Condition Monitoring and Prognosis Method for Incipient Defect Detection and Remaining Life Prediction of Low Speed Slew Bearings
By:Caesarendra, W (Caesarendra, Wahyu)[ 1,2 ] ; Tjahjowidodo, T (Tjahjowidodo, Tegoeh)[ 3 ] ; Kosasih, B (Kosasih, Buyung)[ 1 ] ; Tieu, AK (Tieu, Anh Kiet)[ 1 ]
MACHINES
Volume: 5 Issue: 2
Article Number: 11
DOI: 10.3390/machines5020011
Published: JUN 2017
A Review of Feature Extraction Methods in Vibration-Based Condition Monitoring and Its Application for Degradation Trend Estimation of Low-Speed Slew Bearing
By:Caesarendra, W (Caesarendra, Wahyu)[ 1,2 ] ; Tjahjowidodo, T (Tjahjowidodo, Tegoeh)[ 3 ]
MACHINES
Volume: 5 Issue: 4
Article Number: 21
DOI: 10.3390/machines5040021
Published: DEC 2017
etc.
Author Response
Dear Reviewer:
Thank you very much for your pertinent advice. We have carefully revised the paper again, and finished responses to each point.
Please refer to the attachment for details.
Best wishes

Reviewer 2 Report
The publication entitled “An Optimal Resonant Frequency Band Feature Extraction Method Based on Empirical Wavelet Transform”, which has been sent for review, presents interesting research problem of detect the repetitive transients in the signal .
In reviewer opinion the manuscript should be corrected:
- error in the description - "HZ" change to 'Hz"; "KHz" change to "kHz" !!!
- lack photos of bearing faults
- lack SI units in the figures
- lacks in reference and in the introduction chapter the state of the art of using time and time-frequency signal processing, decomposition of signals and entropy in detect faults of rolling bearings - for example:
- An, X.; Yang, J. Denoising of hydropower unit vibration signal based on variational mode decomposition and approximate entropy. Trans. Inst. Meas. Control 2016, 38, 282–292
- Li, Y.-B.; Xu, M.-Q.; Zhao, H.-Y.; Huang, W.-H. A study on rolling bearing fault diagnosis method based on hierarchical fuzzy entropy and ISVM-BT. J. Vib. Eng. 2016, 29, 184–192
- P. Borghesani, P. Pennacchi, R.B. Randall, N. Sawalhi, R. Ricci: Application of cepstrum pre-whitening for the diagnosis of bearing faults under variable speed conditions, Mechanical Systems and Signal Processing, 36 (2013) 2, 370-384.
- T. Figlus, M. Stanczyk, A method for detecting damage to rolling bearings in toothed gears of processing lines, Metalurgija Volume: 55 Issue: 1 Pages: 75-78 Published: 2016
- J. Antoni, Cyclic spectral analysis of rolling-element bearing signals: facts and fictions, Mech. Syst. Sign. Proc. 304 (2007) 497–529
- Sawalhi N., Randall R.B., 2011, Vibration response of spalled rolling element bearings: Observations, simulations and signal processing techniques to track the spall size, Mechanical Systems and Signal Processing 25 (3), pp. 846-870
- Abboud, Dany; Antoni, Jerome; Eltabach, Mario; et al. Angle\time cyclostationarity for the analysis of rolling element bearing vibrations, MEASUREMENT Volume: 75 Pages: 29-39 Published: NOV 2015
- J Antoni, RB Randall, Differential diagnosis of gear and bearing faults, TRANSACTIONS-AMERICAN SOCIETY OF MECHANICAL ENGINEERS JOURNAL OF VIBRATION AND ACOUSTICS, 2002
Conclusion: this manuscript require the correction and after that should be presented in Entropy.
Regards
Author Response

(The authors gave the same response as above.)

Round 2
Reviewer 1 Report
Figure 8 shoulld be corrected
You have sliding bearing, not ball bearing so figure 8 b) should be redrawn. Remove these 7 circles in the middle.
Author Response
Dear Reviewer:
Thank you very much for your pertinent advice. We have corrected Figure 8 and carefully checked the other Figures.
Best wishes